# Circulating blood extracellular vesicles as a tool to assess endothelial injury and chemotherapy toxicity in adjuvant cancer patients

**Gil Bar-Sela**[1,2]*, **Idan Cohen**[2], **Adva Avisar**[3], **David Loven**[2], **Anat Aharon**[1,4]

1 Bruce Rappaport Faculty of Medicine, Technion-Israel Institute of Technology, Haifa, Israel, 2 Cancer Center, Emek Medical Center, Afula, Israel, 3 University of Haifa, Haifa, Israel, 4 Hematology and Bone Marrow Transplantation, Sourasky Medical Center, Tel Aviv, Israel

\* gil_ba@clalit.org.il

**Data Availability Statement:** All relevant data are within the manuscript and its Supporting Information files.

## Abstract

Extracellular vesicles (EVs) are subcellular membrane blebs that include exosomes and microparticles, which represent a potential source for cancer biomarker discovery. We assess EVs characteristics as a tool to evaluate the endothelial and anti-tumor treatment injury during adjuvant chemotherapy in breast (BC) and colon cancer (CC) patients. Blood samples were taken from 29 BC and 25 CC patients before and after chemotherapy, as well as from healthy control donors (HC). Circulating blood EVs were isolated and characterized by size/concentration, membrane antigens for cell origin, thrombogenicity, and protein content. We observed higher EVs concentration and particle size in CC patients after chemotherapy compared with HC. Higher levels of endothelial EVs (CD144-positive) and vascular endothelial growth factor receptor 1 (VEGFR1), apparently as an indication of endothelial dysfunction, were found in all cancer patients, regardless of a given treatment, compared to HC. Levels of EVs labeled CD62E, CD34$^+$41$^-$, the lymphocyte markers CD11$^+$ and CD-14$^+$, Annexin-V, and the coagulation proteins TF and TFPI, however, sometimes demonstrate significant differences between patients, although HC did not show significant differences between patients pre- and post-chemotherapy. Most importantly, increasing levels of EVs encapsulated Angiostatin were found in patients with CC, while chemotherapy treatment leads to its notable rise in circulating blood EVs. Our results demonstrate the potential of EVs encapsulated Angiostatin as a tool to evaluate endothelial damage during adjuvant chemotherapy in BC and CC patients.

## Introduction

Breast cancer (BC) is the most commonly diagnosed and the second cause of cancer death among women. Colorectal cancer (CC) is the second most common cause of cancer deaths when male and female statistics are combined [1]. Chemotherapy is widely used for both early

**Funding:** This work was funded by the: Israel Science Foundation (ISF), grant No 1413/21, and Rappaport Family Institute Grant 2012-2014. The funders had no role in study design, data collection and analysis, decision to publish, or preparation of the manuscript.

**Competing interests:** The authors have declared that no competing interests exist.

and metastatic diseases to increase the chance of cure, prolong survival, and improve the quality of life in these cancer patients. Nevertheless, chemotherapy-related cellular damage or toxicity remains a significant concern and one of the most central challenges in the delivery of curative cancer therapy [2].

While adjuvant systemic therapies have been shown to have a substantial impact on reducing the risk for breast and colon cancer recurrence and on overall mortality, long-term effects of chemotherapy also significantly increase the risk for secondary leukemia and neuro- and cardio- damage as a result of chemotherapy [3]. Indeed, many of the chemotherapies showed high cardiovascular toxicities and were associated with increased life-threatening cardiovascular complications [4]. Even though the cause of vascular complications after cancer chemotherapy remains mostly ambiguous, it seems that impaired endothelial function, vascular and renal damage, together with oxidative stress and thrombosis, play a significant part in these chemotherapy-related side effects [4, 5].

Extracellular vesicles (EVs) are a varied group of cell-derived membranous structures comprised of exosomes, intracellular luminal vesicles (<100nm), and microparticles (MPs) (100–1000 nm in diameter), which are shed from the cells plasma membranes. They can be found in several biofluids and play a part in numerous physiological and pathological processes [6]. EVs are now considered as a new way of cell-to-cell and tissue communication, enabling the exchange of intracellular molecules, such as DNA, RNA (including non-coding and micro-RNA), and different signaling proteins or lipids [7].

Tissue factor (TF) is considered as a potent pro-coagulant transmembrane protein that initiates the coagulation cascade and blood clotting [8]. TF is abundantly expressed on subendothelial cells, such as fibroblasts, pericytes, and vascular smooth muscle cells, and triggers hemostasis upon vascular injury [9]. Characterization of circulating blood EVs isolated from patients with several different cancer types has shown that EVs express active-surface TF and encapsulate several anticoagulant proteins, such as the TF pathway inhibitor (TFPI) and endothelial protein C receptor (EPCR) with pro-coagulant activity [10]. Moreover, TF-positive EVs were shown to trigger platelet activation *in vivo* and enhanced thrombosis in mice models of venous thrombosis (VT) in a TF-dependent manner [11]. Accordingly, clinically, TF-positive EVs may serve as a useful biomarker and novel therapeutic target to identify patients with cancer at high risk for thrombosis.

Since we recently showed that chemotherapy administration to BC patients affects EVs thrombogenicity and function [12], this study aimed to try and assess the potential of EVs as a tool for managing chemotherapy-related toxicity in patients with breast or colorectal cancer and uncover their cellular origin. We comprehensively compared EVs isolated from healthy donors or breast and colon cancer patients before and after chemotherapy sessions. We assessed their different characteristics to try and evaluate their ability to be used as biomarkers and clinical diagnostic tools for the detection of chemotherapy-associated complications and toxicity.

## Materials and methods

### Study details and ethical approvals

The study with breast cancer patients was approved by the institutional review boards of Rambam Health Care Campus (Approval No. 0368-09-RMB) and HaEmek Medical Center (Israel Ministry of Health Approval No. 920090920). Standardized written consent forms (both in summary and detailed formats) endorsed by the Rambam ethics committee were obtained for each patient in this study and also for the people who gave a blood sample for the control

group. The study with the CC patients was approved by the Institutional Review Board of Rambam Health Care Campus (RHHC) (Approval No. 0153–13).

## EVs isolation and characterization

EVs were isolated, as previously described [12]. In brief, blood samples were collected in tubes containing sodium citrate (3.2%) and EDTA; differential centrifugations were done according to the current gold standard for EV isolation [13]. Specifically, platelet-poor plasma (PPP) was obtained after two sequential centrifugations (15 min 1500g, 24˚C) within an hour of collection and frozen at −80˚C. EVs size, concentration, and membrane antigens levels were validated on PPP samples. In addition, EVs pellets were isolated from thawed PPP by centrifugation (centrifuge Hettich MIKRO 220R, rotor 1189A; Centrifugation condition: 1 hour, 20,000g, 4˚C, braking—0). EV pellets and part of the liquid supernatant fraction were analyzed by protein array. EVs pellets were also analyzed by western blot (WB).

## EVs size and concentration

Size and concentration of PPP EVs (10ul) were evaluated by nanoparticle-tracking analysis (NTA) that can measure particles in the range of 50–2000 nm and concentration in liquids, based on the rate of Brownian movement of nanoparticles in the solution, according to their sizes [14]. Samples measurements were performed using a NanoSight (NTA) version 3.1; Software Version build 3.1.54, Camera Type–sCMOS, Laser Module: NS300, 405nm Software settings for analysis were kept constant for all measurements. Capture Settings: Camera Level: 13; Slider Shutter: 1232; Slider Gain: 219; FPS 25.0; Number of Frames: 749; Temperature: 25 oC, Viscosity: (Water) 0.86 cP; Syringe Pump Speed: 20.

All samples were diluted in 0.025μm filtered PBS to an appropriate concentration before analysis. At least three 30s videos were recorded per sample in light scatter.

## Evaluating EVs membrane antigens

EVs membrane antigen levels were assessed by flow cytometry (FACS-CyAn ADP analyzer, Beckman Coulter). PPP (50ul) were labeled with specific fluorescent antibodies for 30 minutes, at room temperature at dark, ended by adding of 450ul buffer FACS (D-PBS, 0.02% Azid, 0.5% formaldehyde filtered by 0.22μm filter) without additional washing. Fluorescein isothiocyanate (FITC)-Annexin V (Bender MedSystems, Austria) that binds to negatively charged phospholipids as well as with other specific fluorescent antibodies: IgG isotype controls FITC, and PE were purchased from BD Pharmingen, CA. APC IgG1κ Isotype control (R&D Systems, Minneapolis MN), Allophycocyanin (APC)-Flt-1 (vascular endothelial growth factor receptor (VEGFR)-1), PE-KDR (VEGFR-2), Phycoerythrin (PE)-CD41 (platelet marker), PE-anti-CD62E (E-selectin), and anti-CD144 (vascular endothelial cadherin), all purchased from bioLegend, CA. Antibodies for the coagulation markers were: FITC anti-human TF and anti-human TFPI (America Diagnostica, CA). FITC-CD14 (IQP Products, Groningen, Netherlands) and PE-CD11 (BD Bioscience, CA) was used, and leukocyte markers cancer markers: PE-anti-human CD227 (MUC-1). Anti-Epithelial Cell Adhesion Molecule (EpCAM) was purchased from BioLegend. The cell surface glycoprotein A33 and anti-epithelial tumor antigen A33-Alexa flour 488 were purchased from R&D Systems (Minneapolis MN). Anti-mouse IgG-PE (Jackson, PA).

To analyze the size and granularity of "large" EVs by flow cytometer, forward scatter (FSC), and the side scatter (SSC) parameters were set on logarithmic scales as previously described [12]. Briefly, the size and granularity gate were established by standard 0.75μm beads (BD Biosciences, San Jose, CA, USA) (S2A Fig). Filtered PBS (0.22um) sampling was tested to ensure

device cleanliness (S2 Fig). Followed by controls: unstained sample (S2C Fig), and samples labeled by IgG isotype controls FITC/PE/APC (S2D–S2F Fig) to sate the gate of non-specific labeling. For each sample, 10,000 events were count and analyzed by flow cytometers Canto II in log set at the voltage: FSC- 300, SSC-300, FITC-500, PE-350, APC-480. FSC threshold -700. Laser power: blue-20.39; red -12.24; violet-60. Unstained samples and controls were measured in the same setting. In addition, to validate that the particles analyzing by flow cytometer are phospholipid vesicles, detergent (1% tritonx100) was added to 50ul of PPP sample for 5 minutes at room temperature. The CANTO II flow cytometer instrument cannot calculate total EVs concentration; this data obtained only by NTA. EVs protein content measurement.

Protein contents were validated in EVs pellet isolated from thawed PPP by high-speed centrifugation (1 hour, 20,000g, 4˚C) in all study cohorts. Also, protein content was verified in the liquid supernatant fraction (left from the high-speed centrifugation) of the CC group samples and their relative HC group. The samples were screened by the Human Angiogenesis Protein Antibody Array (Ray Bio, Norcross, GA). EVs-pellet or supernatant protein extracts were obtained from a pool of five specimens within each patient sub-group and quantified using the bicinchoninic acid (BCA) protein quantification kit (Thermo Fisher Scientific Inc, IL). Each protein array slide was loaded with 90μg of protein from the EVs pool lysate, and the array was performed according to the manufacturer's instructions. The expression of each protein was represented in quadruplicate on the array slides. Quadruplicate dots identifying each protein were scanned and quantified by GenePIx-4000B (Molecular Devices, Sunnyvale CA). The mean fluorescence intensity of these dots (AU) was determined for each intergroup. Values were compared and normalized to the internal assay controls according to the kit instructions (RayBio Human Cytokine Antibody Array); the amount of biotinylated antibody printed for each Positive Control Spot is consistent from array to array. As such, the intensity of these positive control signals can be used to normalize signal responses for comparison of results across multiple arrays.

## Western blot

EV-pellets that were isolated from 500μl PPP were added to buffer lysis (x2, Ray biotech, or RIPA) with proteinase inhibitor 1% (Sigma). A 50ug sample in buffer lysis was combined with 2 x Laeammil sample buffer contained β-mercaptoethanol (1:20, Biorad). The samples were heated for 5 min 95˚C and loaded and separated on mini protean TGX precast gel 4–20% and then transferred to mini format of transfer blot turbo o.2μm nitrocellulose membrane (Bio-Rad, Herculs CA, USA). This was followed by immunoblotting with rabbit polyclonal anti-human-Angiostatin (Abcam). After overnight incubation with the primary antibody, the membranes were washed and incubated with horseradish peroxidase (HRP) conjugated secondary antibodies (Cell Signaling Technology, Massachusetts, USA). Then, a chemiluminescence kit (WESTAR Nova 2, CYANAGEN, Bologna Italy) was used to detect the fluorescence. The western blot (WB) assay results were quantified using myECL™ Imager and analyzed using MyImageAnalysis Software (both from Thermo Fisher Scientific, Waltham, MA USA).

## Statistical analysis

Data were analyzed using GraphPad-5 software. Continuous variables were reported as mean SD and as median and interquartile ranges. Statistical significance was determined using one-way analysis of variance (ANOVA) followed by Bonferroni's test (*p $<0.05$, **p $<0.01$, ***p $<0.001$). t test-non-parametric Mann–Whitney test was used for comparing two samples. The present study is part of a larger research project testing extracellular vesicles (EVs) properties and different physiological variables in different tumors and treatments. The estimated sample

size for the whole study was calculated according to Cohen's equation; In a regression model to achieve a medium effect size, at a significance level of 0.05, and statistical power of 0.80. However, the results reported in the current study are unplanned subgroup analyses from two different studies that were part of the EVs project.

## Results

### Study cohort

The study with BC patients was conducted between 2009 and 2015. The BC population included patients after surgical removal of the tumor (n = 29), 68% had stage II or III disease, and all patients received adjuvant chemotherapy with four cycles of Adriamycin and Cyclophosphamide given every 2 or 3 weeks, followed by 12 weeks of weekly paclitaxel. The patient's characteristics are summarized in (Table 1). The study with the CC patients was conducted between 2014 and 2016. CC candidates were selected through the hospital's computerized system and included patients diagnosed with CC at stages II-III toward adjuvant chemotherapy following curative surgery. The CC population group (n = 25) included patients after surgical removal of the tumor who were receiving chemotherapy as adjuvant treatment. The median age of patients was 66 years, range 45–83 years; all patients had stage II or III disease and were treated by standard chemotherapy (5-fluorouracil + oxaliplatin or capecitabine + oxaliplatin or capecitabine alone). In this group, most of the patients (n = 18, 72%) were males. Patients characteristics are summarized in (Table 2). Blood samples were obtained from each patient both before the first chemotherapy cycle (time point I) and on the day of the last cycle (time point II). Also, one blood sample was collected from each one of the age-matched healthy controls in both studies. In the study with BC patients (Approval No. 0368-09-RMB), blood samples were collected from 20 healthy women as a control group (median age 51 years, range 32–72 years). In addition, in the study with the CC patients, blood samples were collected from 10 healthy women and five men as a control group (median age 65, range 49–72 years) (Approval No. 0153–13 RHHC). All blood samples were taken with standard 15ml peripheral venous blood sodium citrate (3.2%) tubes. In the breast cancer group, after a median follow-up of four years, only one patient had systemic metastatic disease. In the colon cancer group, after a median follow-up of 2 years, four patients had recurrent metastatic disease. All patients were alive at the end of the study follow-up period. The small numbers of patients with a recurrent metastatic disease did not allow any conclusion on the correlation between EV characteristics and disease prognosis.

### EVs concentration and size distribution

To try to assess EVs characteristics as a tool to evaluate endothelial injury and anti-tumor treatment complications during adjuvant chemotherapy in breast (BC) and colon cancer (CC), we first analyzed concentration and size distribution of EVs in PPP isolated from patients' blood. Representative histograms showing the size distributions of each patient's subgroup presented in (S1 Fig). Samples obtained from CC patients before (CCI) and at the end of chemotherapy treatment (CCII) were significantly more concentrated than EVs obtained from healthy controls (HC) (CCI- 1.2883e+009±9.9160e+008, $p$ = 0.0317 t-test, CCII—1.7083e +009±2.1334e+008, $p$ = 0.0159 t-test compared to HC—4.5602e+008±1.5489e+008) (Fig 1A). Moreover, EVs obtained at the end of chemotherapy from CCII patients were much larger, 154.0±10.42 nm, than HC EVs (88.63±28.60nm, $p$<0.01 Anova test) or compared to EVs obtained from BC patients at the end of treatment (BCII-108.8±17.15nm) and include fewer small EVs (under 100nm) than BCII or HC (Fig 1B and 1C).

**Table 1. Breast cancer patient's characteristics.**

| | | Number of patients (%) |
|---|---|---|
| **Age** | <40 years | 4(14%) |
| | >40 years | 25(86%) |
| **Medical history of VTE*** | | 2(7%) |
| **Stage** | IA or IB | 10(34%) |
| | IIA | 9(31%) |
| | IIB | 4(14%) |
| | IIIA | 2(7%) |
| | IIIB or IIIC | 4(14%) |
| **Grade** | I | 0 |
| | II | 10(34%) |
| | III | 18(62%) |
| | Unknown | 1(4%) |
| **Tumor size** | <2 cm | 12(41%) |
| | 2-5cm | 15(52%) |
| | >5 cm | 2(7%) |
| **Number of lymph nodes involved** | 0 | 12(41%) |
| | 1–3 | 13(45%) |
| | 4–9 | 3(10%) |
| | 10+ | 1(4%) |
| **Estrogen receptor (ER)** | Negative | 13(45%) |
| | Positive | 16(55%) |
| **Progesterone receptor (PR)** | Negative | 11(38%) |
| | Positive | 18(62%) |
| **Triple negative breast cancer** | | 9(31%) |
| **Adriamycin and cyclophosphamide** | Given every 3 weeks | 7(24%) |
| | Given every 2 weeks (dose dense) | 18(62%) |
| | | 4(14%) |
| | Not given | |
| **Taxane base chemotherapy** | Not given | 7(24%) |
| | Paclitaxel weekly for 12 weeks | 14(48%) |
| | Paclitaxel every 2 weeks (dose dense) | 8(28%) |
| | Docetaxel-carboplatin-Trastuzumab | 4(14%) |
| **Operation** | Lumpectomy+SLNB** | 8(28%) |
| | Lumpectomy+ALND*** | 9(30%) |
| | Mastectomy+SLNB | 4(14%) |
| | Mastectomy+ALND | 8(28%) |
| **Trastuzumab** | No | 22(76%) |
| | Yes | 7(34%) |

## EVs cell origin

The majority of EVs measured by the flow cytometer found to be phospholipid vesicles located in the same area where 0.75μm beads are located (R6, S1A Fig), as confirmed by the treatment of PPP with 1% triton-x100 that reduced EVs number in this area by above ten times. (S3 Fig). Expression of EVs cancer markers, such as MUC1-CD277, a transmembrane protein involved in metastasis, A33 a glycoprotein found on most primary and metastatic colorectal cancers, and epithelial cell adhesion molecule (EPCAM), a transmembrane glycoprotein involved in

**Table 2. Colon cancer patient's characteristics.**

| | Number of patients (%) |
|---|---|
| **Age, years** | 66 |
| Median (min-max) | (45–83) |
| **Males, n (%)** | 18 (72%) |
| **Females, n(%)** | 7 (28%) |
| **Cancer disease:** | |
| stage 2 | 2 (8%) |
| stage 3 | 23 (92%) |
| **Chemotherapy:** n (%) | |
| Capecitabine | 3 (12%) |
| Apecitabine, Oxaliplatin | 18 (72%) |
| 5-fluorouracil, Leucovorin, Oxaliplatin | 4 (16%) |
| **Other parameters:** | |
| Heart disease, n (%) | 1 (4) |
| Diabetes, n (%) | 6 (24) |
| Hypertension, n (%) | 9 (36) |
| Hyperlipidemia, n (%) | 3 (12) |
| Anticoagulants, n (%) | 7 (28) |

tumorigenesis, were explored. T-test analysis of MUC1-CD277 displayed higher expression in BCI patients before treatment (BT) than in CCI patients BT ($p = 0.0012$). A33 EVs expression demonstrated significantly higher levels in CCI ($p = 0.0047$) and CCII (0.045) compared to relative HC (Fig 2). The levels of RBC EVs were found to be similar in the study cohorts while

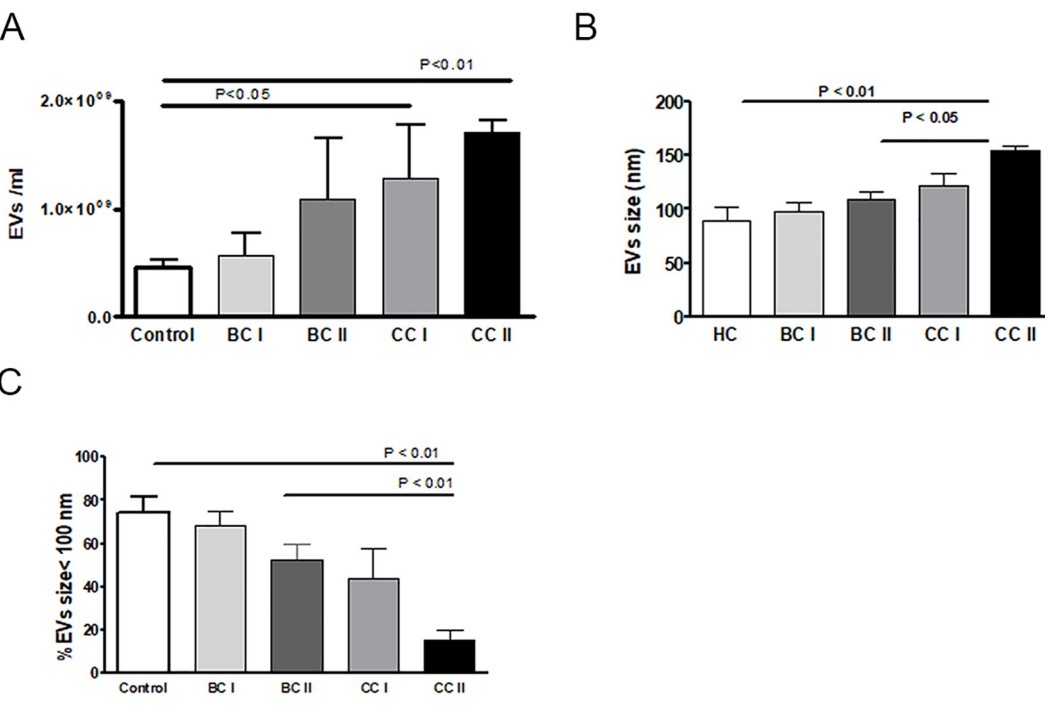

**Fig 1. EVs size distribution and concentration.** EVs were isolated as described in materials and methods (under section EVs isolation and characterization). (A) Concentration or (B and C) size distribution of EVs were determined by nanoparticle tracking analysis (NTA).

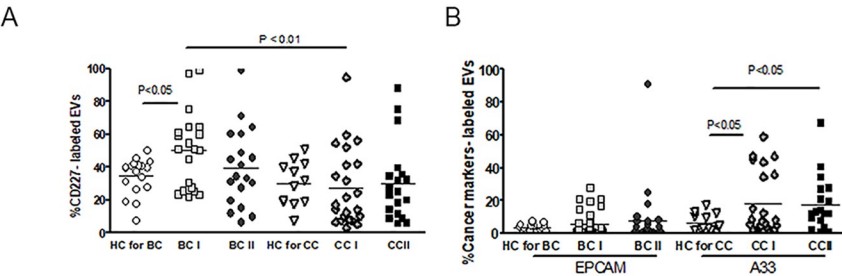

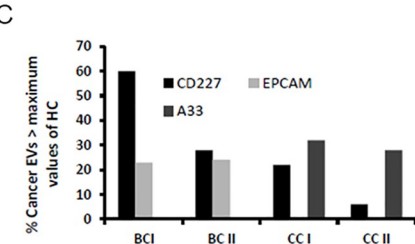

**Fig 2. EVs cancer markers.** The percentage of EVs labeled with (A) MUC1-CD277 and (B) EPCAM (left) or A33 (right) were measured using FACS with specific fluorescent antibodies. The percentages of labeled EVs were calculated from the total EVs number. (C) Percentage of cancer patients EVs that express cancer markers, above the maximum values that were found in EVs of HC.

significantly decreased as documented in platelet EVs in BC patients and a trend of decrease in CC patients compared to HC (S1 Table).

Although the role of endothelial cell (EC) dysfunction and damage in the regulation of vascular disease is well-known [15], their contribution to the regulation of tumor progression and chemotherapy-related complications in cancer patients is not well studied. To try to assess endothelial cells damage and their possible contribution to vascular complications during cancer chemotherapy, we comprehensively characterized circulating blood EV from healthy donors or BC and CC patients before and after chemotherapy sessions. We first examined the level of CD144 (vascular endothelial cadherin, VE-cadherin), a marker for endothelial gap junction, in circulating blood EVs, as disease-related endothelial dysfunction and a damage and chemotherapy-related complications monitoring tool. The levels of CD144-EVs (Fig 3A) demonstrated significantly higher levels in both subgroups of BC and CC BT (BCI, CCI) and at the end of standard chemotherapy treatment (BCII, CCII) than in their relative healthy control (HC) groups, correspondingly. However, no significant elevation in the levels of CD144-EVs could be observed between patients before and after treatment (group BCI to BCII and CCI to CCII) (S1 Table, Fig 3A).

We further tested the levels of EVs expressing CD62 antigen-like family member E (CD62E), a marker for endothelial activation (expressed on endothelial cells activated by cytokines), in circulating blood EVs isolated from the same groups. The level of CD62E on endothelial cells EVs (EC-EVs) was higher in BC patients before and after chemotherapy compared to EVs obtained from their relative healthy control (over three times mean levels) but did not change during treatment (groups BCI to BCII) (Fig 3B). Unexpectedly, we did not observe any significant parallel difference in mean levels of CD62E positive EC-EVs between CC patients compared to the levels of the HC group. The levels of CD62E were significantly higher in the BC compared to CC patients BT ($p<0.001$) and after chemotherapy ($p<0.01$).

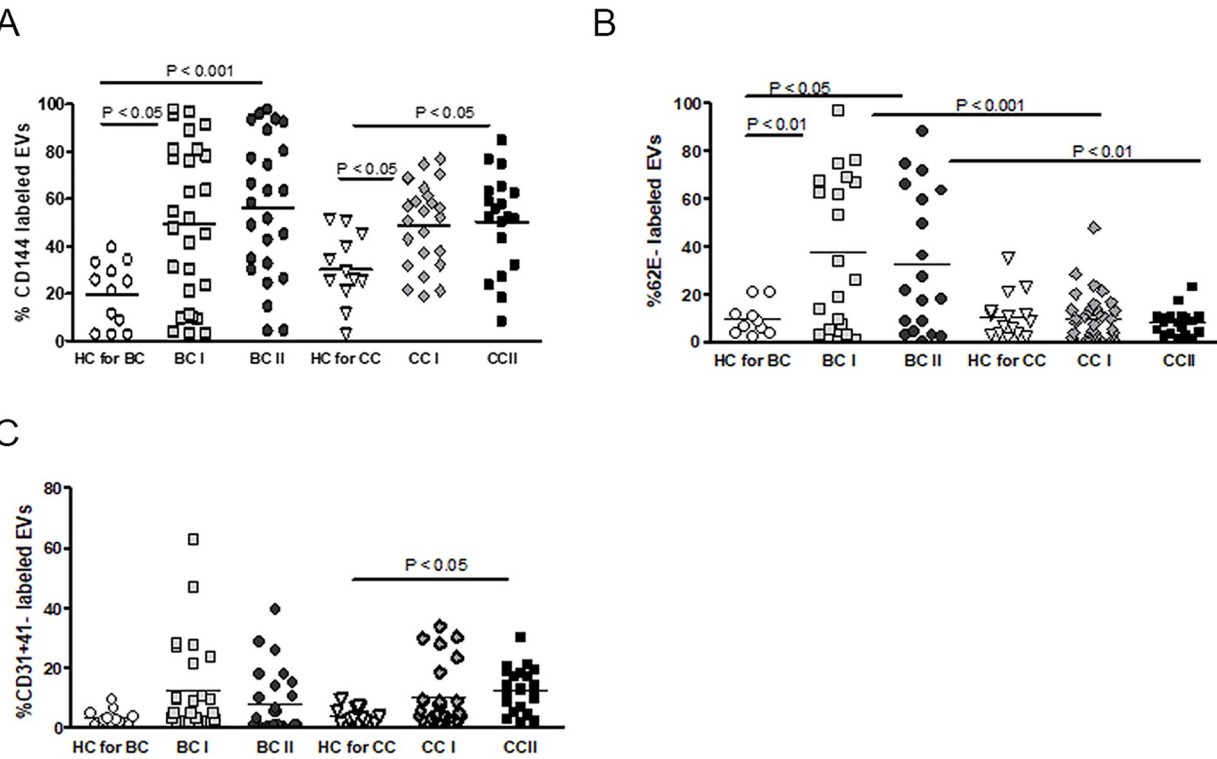

**Fig 3. EVs endothelial markers.** Antigen levels of (A) VE-cadherin (CD144), (B) endothelial markers and E-selectin (CD62E), and (C) CD31$^+$/ CD41$^-$ expression was measured with FACS using specific fluorescent antibodies on EVs obtained from HC or patients before chemotherapy (time point I) and at the last chemotherapy treatment (time point II). The percentages of labeled EVs were calculated from the total number of EVs.

We then also tested CD31$^+$/CD41$^-$ (CD31$^-$ PECAM; platelet endothelial cell adhesion molecule, CD41$^-$ Integrin alpha-IIb; expressed in platelets) expression on EVs obtained from all HC and patient sub-groups. Contrary to the trend observed for CD62E, the levels of the endothelial markers CD31$^+$/CD41$^-$ were > two times higher in the CC subgroup compared to those obtained from their relative HC group (Fig 3C) and remained high after standard chemotherapy without significant difference from levels in CCI. However, CD31$^+$/CD41$^-$ EC-EVs levels showed no significant change during the treatment of BC patients (BCI to BCII) and compared to HC (Fig 3C).

Next, we tested the possibility that white blood cell EVs may better reflect drug toxicity and tissue damage after chemotherapy resulting from sterile inflammation and immune cell activation by dying cells and damage-associated molecular patterns (DAMPs) signaling [16]. The levels of leukocyte-derived EVs, labeled with CD11, were found to be significantly higher in BC patients before and after chemotherapy compared to their relative HC group (BCI, $p<0.01$, BC II, $p<0.05$) (Fig 4A). EVs isolated from CC patients did not show any difference in mean levels of CD11 to their relative HC group. Levels of CD11 EVs in BC patients BT were much higher than in CC patients BT ($p<0.001$). Additionally, measuring myeloid cell-specific leucine-rich glycoprotein (CD14) expressing EVs levels did not show any specific trend for monocyte-derived EVs, neither in cancer patients nor after chemotherapy treatment. However, CD14 EVs levels were significantly lower in CC patients before chemotherapy (CC-I) ($p<0.05$) (Fig 4B).

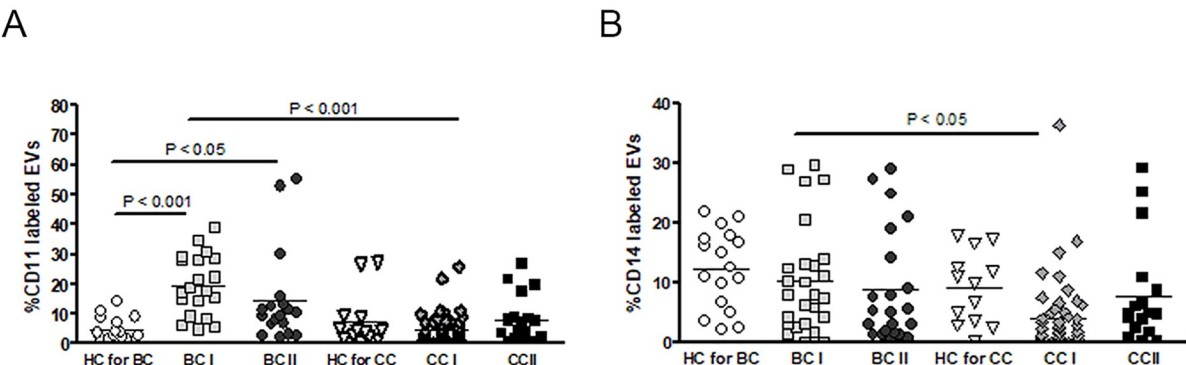

**Fig 4. EVs leukocyte markers.** Antigen levels of (A) CD-11 or (B) CD-14 were measured on EVs isolated from HC and BC or CC patients using specific fluorescent antibodies, as described in Fig 1. The percentages of labeled EVs were calculated from the total number of EVs.

### Characterization of EVs coagulation proteins

Annexin-V positive EVs were previously associated with systemic inflammation response in patients with traumatic injuries [17], while cancer phosphatidylserine-containing EVs were shown to be critical for coagulation activation [18, 19]. Either way, it seems that the release of Annexin-V positive EVs from tissues or solid tumors are mostly dependent on the cell type and stimulation conditions, which may be affected by chemotherapy [20]. We then measured the expression of negatively charged phospholipids on EVs membrane surface using Annexin-V binding. No significant differences were found between all patients' subgroups to their relative HC (S1 Table).

Next, we measured the level of pro- and anticoagulant proteins (Table 3 and S1 Table). A significant difference in the pro-coagulant protein tissue factor (TF) was found between both BC patients subgroups (BC I, $p<0.05$; BC II, $p<0.05$) and both CC patients' subgroups (CC I $p<0.01$; CC II $p<0.001$) to their HC (Fig 5A). Testing the levels of TF pathway inhibitor (TFPI), a kunitz-type serine protease inhibitor that regulates the TF-dependent pathway of blood coagulation-bearing EVs, showed only a trend of decrease in patients' subgroups TFPI mean levels compared with their relative HC levels (S1 Table). We then decided to calculate the ratio of TF/TFPI encapsulated in circulating EVs to test the possibility that it may affect the patient's coagulation potential and propensity to activate the coagulation cascade. We could not observe significant differences; only a trend of the increase was found in EVs TF/TFPI ratios of all cancer patients' subgroups (before or after treatment) to their relative HC. However, >30% of BC patients and >19% of CC patients demonstrated EVs TF/TFPI ratios above the maximum value of relative HC (S1 Table).

We also tested the levels of endothelial protein C receptor (EPCR, or CD201), a receptor for protein C, which is activated by the blood coagulation pathway. This receptor is known as an anticoagulant, while its soluble form (sEPCR) counts as a pro-coagulant that inhibits protein C activation and blocks APC anticoagulant function [21]. The levels of EPCR-E's were 3.5–4.8 times higher in all cancer subgroups (BCI and BCII or CCI and CCII) compared to their relative HC without any observed difference between cancer subgroups before and after treatment (BCI to BCII and CCI to CC II) (Fig 5B).

### Vascular endothelial cell growth factor incorporated EVs

The expression of vascular endothelial growth factor receptors (VEGFR1/FLT1 and VEGFR-2/KDR) is typically found in vascular endothelial cells, placental trophoblast cells, and

**Table 3. Summary data of all blood circulating breast and colon cancer EVs examined factors and surface proteins.** Results are expressed as change in expression relative to HC.

|  | BC-I | BC-II | CC-I | CC-II |
|---|---|---|---|---|
| CD144 | ▲ | ▲ | ▲ | ▲ |
| CD62E | ▲ | ▲ | – | – |
| CD34$^+$41$^-$ | – | – | ▲ | ▲ |
| CD11 | ▲ | ▲ | – | – |
| CD14 | – | – | – | – |
| Annexin-V | – | – | – | – |
| TF | ▲ | – | ▲ | ▲ |
| TFPI | – | – | – | – |
| EPCR (CD201) | ▲ | ▲ | ▲ | ▲ |
| VEGFR1 | ▲ | ▲ | ▲ | ▲ |
| VEGFR2 | ▲ | ▲ | ▲ | ▲ |
| Angiopoietin-1 | – | – | – | – |
| Angiostatin | – | – | ▲ | ▲* |
| EGF | – | – | – | – |
| G-CSF | – | – | ▲ | – |
| GM-CSF | – | – | ▲ | – |
| IL-10 | – | – | ▲ | – |
| I-TAC (CXCL11) |  |  |  |  |
| MMP-9 | – | – | – | – |
| PECAM-1 | ▼ | – | ▲ | – |
| uPAR | – | – | – | – |
| VEGF | ▲ | – | ▼ | ▼ |

* Statistically significant elevation relative to CC-I group.

peripheral blood monocytes [22]. Since both solid tumors and blood vessels were shown to regulate carcinogenesis, angiogenesis, and endothelial cells growth and survival by VEGF and its receptors [23], we measured the levels of FLT1 and KDR on the surface of EVs obtained from our patients, as well as their relative HC. The levels of FLT1 and KDR on the surface of EVs obtained from HC were low (S1 Table). They were found to be more than 3-fold higher in

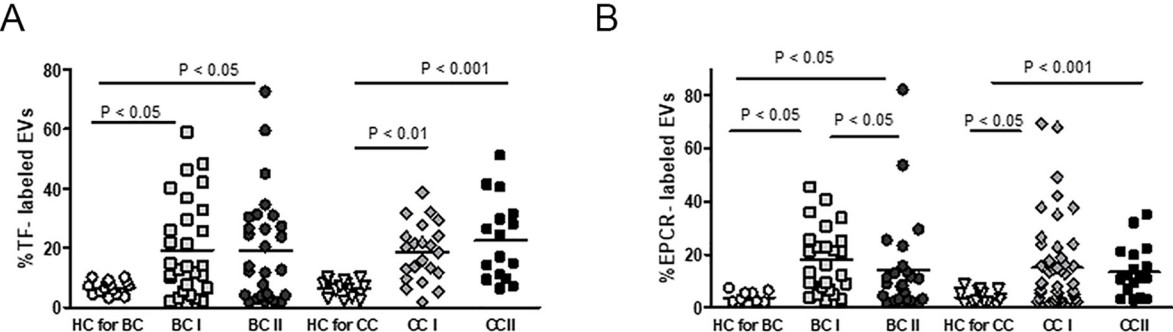

**Fig 5. EVs thrombogenicity.** (A) The levels of coagulation marker TF and TFPI (see Table 1) were measured on EVs obtained from BC and CC patients and their relative HC using specific fluorescent antibodies. The percentages of labeled EVs were calculated from the total number of EVs using FACS analysis. (B) The mean levels of EPCR on HC and cancer patients EVs were measured as described previously (in Fig 2).

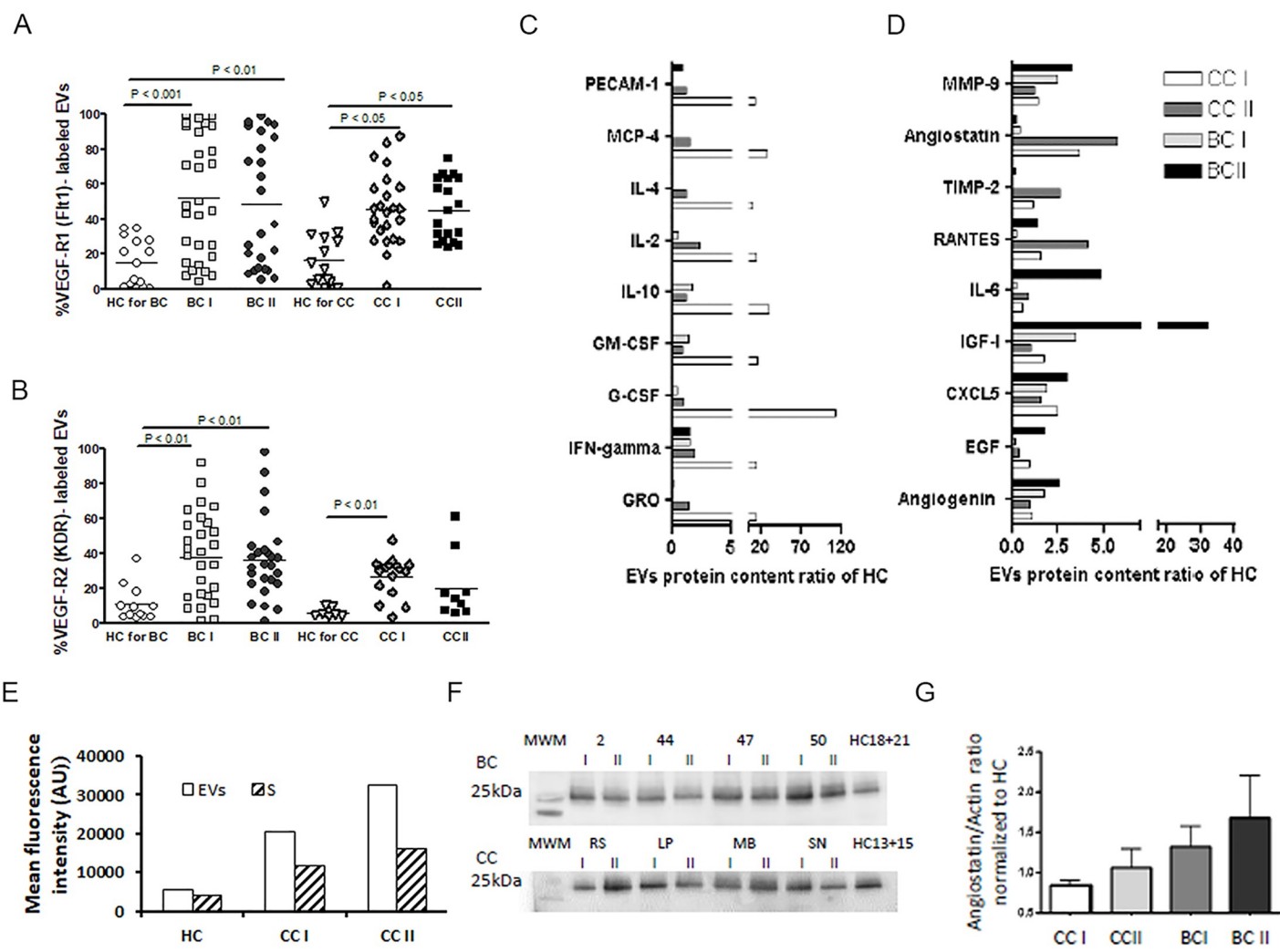

**Fig 6. EVs growth factor receptors expression.** Levels of the growth factors receptors (A) VEGFR1 (FLT1) and (B) VEGFR-2 (KDR) were measured on EVs isolated from HC and BC or CC patients before chemotherapy (time point I) and at the last chemotherapy treatment (time point II) using specific fluorescent antibodies (as described in Fig 2). The percentages of labeled EVs were calculated from the total number of EVs. (C-E) Measuring EVs encapsulated angiogenic related proteins. EVs protein extracts were obtained from a pool of five specimens within each patient subgroup and validated by Human Angiogenesis Protein Antibody Array. Slides were analyzed using TotalLab software results. Each protein has significant signal intensities representing protein content, expressed in arbitrary units (AU). For simplicity, the changes in protein contents are normalized and presented as AU relative HC (C and D) when (E) shows Angiostatin in EVs pellet (EVs) and the supernatant liquid fraction marked as (S) of HC, CC I, and CCII patients. (F) WB gel images of Angiostatin content in EVs pellet obtained from BC patients (n = 4) and CC patients (n = 4) in each time point (I, II) and in EVs obtained from HC. (G) Graph F summaries the WB results as average with STD of EVs Angiostatin as a ratio of Actin, normalized to HC.

both BC ($p<0.01$) and CC ($p<0.05$) patient groups but remained similarly high in the BCII and CCII groups following chemotherapy treatment (Fig 6A and 6B). Hence, it seems that the high levels of VEGF-R-positive circulating EVs most likely reflect their expression on solid tumors or activated leukocytes and, to a lesser extent, the damaged endothelial vascular cells in post-chemotherapy treatments.

## EVs encapsulated protein loads characterization

To study the protein profile of circulating EVs-associated with chemotherapy toxicity, we used a human angiogenesis antibodies array screen for angiogenesis-relegated enzymes, inhibitors,

cytokines, or different growth factors and proteins. We used EVs pellet samples from two different cancer patients cohorts (e.g., CC and BC) and two HC control groups and the supernatant liquid left from the CC group and their relative HC. We found that, of the 60 different proteins that were checked, their content as packed in EVs pellet fraction was found to be significantly higher compared to their contents in the supernatant liquid fraction (EVs/ supernatant >1.3). Specifically, 77% of proteins of the CCI sample, 60% of the proteins of the CC II sample, and 68% of proteins in the HC sample were found to be more concentrated in EVs pellets than in the supernatant. In EVs pellets, a designated group of proteins, including GRO, IFN-gamma, G-CSF, GM-CSF, IL-10, Il-2, Il-4 MCP-4, and PECAM-1, showed a dramatic (above 10-times relative to HC) and specific elevation in the means levels only in CC1 patients. In these patients, their levels were reduced at CC II, similar to HC (Fig 6C). Interestingly, most of those proteins were previously detected and validated in multiple targeted proteomics cancer biomarkers studies [24]. There are specific proteins whose contents significantly increased post-chemotherapy in EVs of CCII and others in EVs samples of BC II patients (Fig 6C). In the BC patients, levels of IGF-I and IL-6 increased more than 5 times post-chemotherapy (Fig 6C). In contrast, in CC patients, the endogenous angiogenesis inhibitor Angiostatin showed a specific elevation in the CCI patient group, and its levels were sharply intensified after chemotherapy treatment (Fig 6D). Comparing proteins content in the EVs and the supernatant fraction demonstrated that the majority of Angiostatin is located in EVs fraction, and its levels are significantly higher in the patient samples compared to HC (Fig 6E). The additive increase found in CC II compared to CC I is related mainly to the protein accumulation in the EVs fraction. WB analysis supports these findings; a non-significant increase in Angistatin was found in EVs pellets obtained from patients at the end of chemotherapy in both diseases (CC II, BC II) compared to their levels before treatment (CC II, BC I); (Fig 6F and 6G). These results suggest that EVs encapsulated Angiostatin has the potential to be used as a CC diagnostic marker for chemotherapy-associated vascular injury and associated complications.

## Discussion

EVs play a central role in the early pathogenesis and subsequent progression of many human diseases. Higher plasma circulating EVs originating from leukocytes, red blood cells, platelets, or other organs and tissues have been reported in many types of cancers [25], cardiovascular and metabolic diseases [26, 27], and even neurological pathologies, such as Multiple Sclerosis (MS) and Alzheimer's disease [28]. This suggests that circulating EVs may transport common informative cellular signals in pathophysiological conditions on vital cellular and physiological processes, such as cell activation or injury, and could serve as a potential biomarker for disease diagnosis or therapeutic monitoring.

Our study aimed to harness the strength of circulating blood EV membrane receptors and encapsulate molecules for the detection of diagnostic and monitoring tools to assess the rates of inflammation, angiogenesis, and potential for thrombosis or vascular damage in BC and CC patients before and after chemotherapy. Therefore, EVs size, concentration, and membrane antigen were studied on PPP. Moreover, in protein screening in EVs pellet (obtained by high centrifugation versus the supernatant liquid fraction), we found that the majority of proteins are packed in EVs rather free in the plasma.

Indeed, previous studies have shown that circulating EVs take part in the coagulation process when endothelial cells, platelet, and monocyte-derived EVs promote the assembly of the enzyme complexes acting on the coagulation cascade, resulting in cell fusion events that may lead to thrombus formation [10, 29]. Circulating EVs were also shown to promote anti-inflammatory or pro-inflammatory processes when their interaction with the target cell leads to a

robust secretion of cytokines [30]. Therefore, we decided to comprehensively study, catalog, and compare membrane proteins and intracellular factors of circulating EVs taken from BC and CC patients pre- or post-chemotherapy sessions and to compare them to their relative HC groups, to try to uncover common essential elements between the two malignancies.

Overall, in our comprehensive surveys (summarizing the overall differences between HC control groups and the relevant cancers, (Table 1)), it seems that most of the substantial differences we observed between datasets of BC or CC patients compared to their relative control groups in EVs encapsulated proteins may be primarily influenced by the different type of the probed malignancy. These differences were mainly derived from disease-related factors of the examined cancers (e.g., the disease's effect on the levels of leukocytes) that could not be attributed to the kind of given chemotherapy. In our control groups, we used two different (non-parallel) groups of subjects (varied in age and gender) and could not find any significant differences between the two HC groups. Therefore, we suggest that circulating EV particles can provide a reliable clinical tool in cancer diagnosis and may be a highly suitable method to assess cancer therapy effectiveness or risk assessment, as it enables comparison between different groups of cancer patients. Nevertheless, it seems that the high variance between proteins in cancer EVs among various cancers resembles the one that can be seen in proteins discovered in the serum. This allows EVs, at least in this aspect, no additional advantage over blood or plasma biomarker proteomics at this time.

Cancer-associated thrombosis is a well-known phenomenon and a common complication in cancer patients with relative risks (RRs) 4–7 of venous thrombosis compared with the general population or patients without cancer [31]. EVs express coagulation factors and play a central role in promoting thromboembolic events [32]. The high thrombogenic profile of EVs found in adjuvant patients may indicate that EVs thrombogenicity is not related only to tumor cells but is probably associated with endothelial injury and an increase of inflammation. The type of cancer affects the incidence rates of VT [33], and a higher incidence of VT (per 1000 person-years) was found in colorectal patients than in breast cancer patients. However, in the current study, EVs thrombogenicity, as reflected in the levels of EVs: TF, TF/TFPI ratio, and EVs EPCR (count as soluble EPCR, a pro-thrombotic marker) were found to be similar in breast cancer and colorectal patients.

Interestingly, in contrast to our initial assumption and based on previous studies analyzing several EV membrane markers, such as endothelial injury markers (CD144, CD62E, CD34[+]41) and VEGFR1/2, white blood cells markers (CD11[+] and CD14[+]), or the coagulation proteins (TF and TFPI and EPCR), we found no distinct pattern between patients pre- and post-chemotherapy in any of the cancer patients. Either way, the strength of our work is validated by the fact that we could observe elevation in previously reported cancer-specific biomarkers in specific cancers, such as G-CSF, GM-CSF, IL-10, and PECAM-1 (**see** Table 1 for HC to cancer differences) [24]. However, when using specific angiogenic proteins array, to both fractions EVs pellet and supernatant liquid, we were able to demonstrate a gradual elevation in Angiostatin (five times higher relative to HC) with chemotherapy treatment in CC-II patients mainly encapsulated in the EVs fraction (Fig 6). Direct Angiostatin treatment is known to be cytotoxic to tumor cells (at low extracellular pH in a surface-associated ATP synthase dependent mechanism) [34] but, most importantly, was found to be a potent inducer of apoptosis in endothelial cells [35]. Moreover, Angiostatin binds to ATP synthase on the surface of human endothelial cells and inhibits endothelial cell migration, proliferation, and tube formation [36]. It is proposed that Angiostatin might disrupt the production of ATP and thus render endothelial cells susceptible to hypoxic damage [37]. Treatment of endothelial cells with the antioxidant N-acetylcysteine abrogates morphological changes and cytotoxic effects of Angiostatin treatment, supporting the model in which Angiostatin induces a transient rise

in free radical production [38]. Accordingly, high levels of Angiostatin EVs, as a result of prolonged chemotherapy treatments, may lead to substantial endothelial dysfunction, diminished angiogenic wound repair, and excess inflammation.

Altogether, in this study, we show a sizeable comprehensive mapping of EVs-related intracellular markers for BC and CC chemotherapy toxicity, focusing on endothelial injury and thrombogenicity. Our results, although additional studies are required, demonstrate the potential of Angiostatin as a chemotherapy toxicity monitoring tool, which may further be considered in the following studies as a therapeutic target for preventing chemotherapy vascular or angiogenic cancer complications.

## Supporting information

**S1 Fig. EVs size distribution diagram of a representative sample obtained from each study cohort.** The graphs summarized 3–4 repeats of analysis for each sample and expressed as Averaged FTLA Concentration / Size for HC-EVs (A), CC I-EVs (B), CC II-EVs (C), BC I-EVs (D), and BC II-EVs (E).
(TIF)

**S2 Fig. FACS EVs setting and controls.** (A) Estimate gate for 0.75 um beads (r1) used for EVs size evaluation. (B) Filtered (0.22um) PBS. (C) Unstained sample PPP-EVs distribution. Forward Scatter (FSC) information about particle size vs. Side Scatter (SSC) information about particles granularity. EVs samples were labeled with IgG isotype controls; the graph presented the fluorescent intensity. The gate was set for positive label area, (D) IgG isotype controls FITC (E) IgG isotype controls PE, and (F) IgG isotype controls APC.
(TIF)

**S3 Fig. FACS analysis of PPP before and after triton treatment.** Unstained PPP EVs are located in the same area were 0.75μm beads are located (R6, S1A Fig). (A). HC11–79% EVs located at R6 (B). BC2 II: 92% EVs located at R6. Treatment with 1% triton-x100 reduced EVs number in the R6 area by above ten times. (C). HC 11: EVs located at R6, decreased to 6.6%. (D). BC2-II: EVs, located at R6, reduced to 8.8%.
(TIF)

**S1 Table.**
(PDF)

## Acknowledgments

The authors would like to thank all patients, families, and caregivers who participated in the study.

## Author Contributions

**Data curation:** Gil Bar-Sela.

**Formal analysis:** Gil Bar-Sela, Idan Cohen, Adva Avisar, Anat Aharon.

**Funding acquisition:** Anat Aharon.

**Investigation:** Gil Bar-Sela, Adva Avisar, David Loven.

**Methodology:** David Loven.

**Supervision:** Gil Bar-Sela, Anat Aharon.

**Writing – original draft:** Gil Bar-Sela, Idan Cohen, Anat Aharon.

**Writing – review & editing:** Idan Cohen, Anat Aharon.

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
