## [Decision Letter · Decision Letter 0]

9 Jun 2020

PONE-D-20-06807

Circulating Blood Extracellular Vesicles as a Tool to Assess Endothelial Injury and Chemotherapy Toxicity in Adjuvant Cancer Patients

PLOS ONE

Dear Dr. Bar-Sela,

Thank you for submitting your manuscript to PLOS ONE. After careful consideration, we feel that it has merit but does not fully meet PLOS ONE’s publication criteria as it currently stands. Therefore, we invite you to submit a revised version of the manuscript that addresses the points raised during the review process.

Please address the very serious concerns raised by the reviewers.

We look forward to receiving your revised manuscript.

Kind regards,

Jeffrey Chalmers, Ph.D.

Academic Editor

PLOS ONE

Journal Requirements:

3. Please provide additional details regarding participant consent for the colon cancer population group. In the ethics statement in the Methods and online submission information, please ensure that you have specified (a) whether consent was informed and (b) what type you obtained (for instance, written or verbal, and if verbal, how it was documented and witnessed). If your study included minors, state whether you obtained consent from parents or guardians. If the need for consent was waived by the ethics committee, please include this information.”

4. In your Methods section, please provide additional information about the participant recruitment method and the demographic details of your participants. Please ensure you have provided sufficient details to replicate the analyses such as: a) the recruitment date range (month and year), b) a description of any inclusion/exclusion criteria that were applied to participant recruitment, c) a table of relevant demographic details, d) a description of how participants were recruited, and e) descriptions of where participants were recruited and where the research took place."

5. Please provide a sample size and power calculation in the Methods, or discuss the reasons for not performing one before study initiation."

6. In the Methods section, please provide the source, product number and any lot numbers of the antibodies purchased for your study

7. Please include your tables as part of your main manuscript and remove the individual files. Please note that supplementary tables (should remain/ be uploaded) as separate "supporting information" files

Reviewers' comments:

Reviewer's Responses to Questions

**Comments to the Author**

1. Is the manuscript technically sound, and do the data support the conclusions?

Reviewer #1: No

Reviewer #2: Yes

2. Has the statistical analysis been performed appropriately and rigorously? 

Reviewer #1: No

Reviewer #2: Yes

3. Have the authors made all data underlying the findings in their manuscript fully available?

Reviewer #1: No

Reviewer #2: Yes

4. Is the manuscript presented in an intelligible fashion and written in standard English?

Reviewer #1: Yes

Reviewer #2: Yes

5. Review Comments to the Author

Reviewer #1: The manuscript by Bar-Sela describes the analysis of citrated plasma from breast cancer patients for extracellular vesicles (EVs). The authors fractionate citrated plasma by centrifugation and analyze the resulting pellet using NTA, flow cytometry, and antibody array. They report differences in the fluorescence signals produced by samples stained with different antibodies, and an increase abundance of angiotensin is lysed material. The authors propose that angiostatin-bearing EVs are a useful biomarker for endothelial damage.

The authors contentions are hard to evaluate from the available data and impossible to reproduce given the information provided.

The authors have chosen to focus on a pellet of material obtained after centrifugation of previously-frozen citrated plasma for 1 hour at 20,000xg. This is a very crude fractionation and the pellet obtained will likely contain some EVs plus other sediments, but will leave significant material in the supernatant. The authors should justify their choice of fractionation method, and describe it more completely including information about starting and final volumes, concentration factor, and the centrifuge, rotor, and any braking, etc.

The authors state that EV size and concentration was evaluated by NTA, but present only a cursory description and results. The authors should present representative histograms to show the size distributions, and relevant details about the measurement, including the length of the tracking videos, the analysis algorithm used to analyze the data, and variance in their triplicate technical replicates. How did the authors evaluate the fraction of non-EV particle in their sample?

The authors use flow cytometry analyze their samples, but the data is uninterpretable from the information provided. The authors are referred to the guidelines on reporting EV flow cytometry methods and results recently published in the Journal of Extracellular Vesicles, which also includes information on essential controls and calibration.

Finally, the authors claim the angiostatin carried inside EVs is a useful potential biomarker, but the mere presence of angiostatin in a crude pellet that also contains some vesicles is unconvincing. Stronger evidence of the vesicular nature of might come from a cleaner EV prep that had been further fractionated to remove non-EV contaminants, and showing that the angiostatin is lost from the vesicle fraction after treat met with detergent, for example.

Reviewer #2: 1. Abstract: Anti tumor treatment toxicity typically connotes organ toxicities or symptom burden experience by patients. If not patient reported outcomes or actually toxicities are reported, I suggest that this not be described in this way since it is very misleading

2. Introduction: 2nd sentence in introduction is a run off sentence and needs to be edited. As stated above, I disagree with the firm association of chemotherapy toxicities with EVs. The authors have no strong data to assert this. the findings of EV in these 2 patient cohorts can be certainly described and it's very interesting. and in the discussion, conjectures can be made with chemotherapy associated complications and toxicity but the study was not poised to answer this question.

3. Methods: please move discussion of patient demographics and tumor characteristics to results section. Describe how healthy controls were obtained and under what human protocol. Was it under patient IRBs?

4. Results: please start with describing patient clinical demographics. Given long term nature of the data, are the authors interested in reporting disease outcomes in these 2 cohorts-- Since pts who had earlier recurrences may have influenced EV characteristics. If not able to do this, please acknowledge this as a limitation. comments above apply to discussion of results. We cannot draw conclusions on toxicity based on these results.

5. discussion of thrombosis factors but without reporting whether pts had thrombotic events needs to put in better context. Not enough data for the conclusions listed

6. PLOS authors have the option to publish the peer review history of their article (what does this mean?). If published, this will include your full peer review and any attached files.

Reviewer #1: No

Reviewer #2: No

---

## [Author Response · Author response to Decision Letter 0]

2 Jul 2020

We added all necessary information in the new revised manuscript

---

## [Decision Letter · Decision Letter 1]

31 Jul 2020

PONE-D-20-06807R1

Circulating Blood Extracellular Vesicles as a Tool to Assess Endothelial Injury and Chemotherapy Toxicity in Adjuvant Cancer Patients

PLOS ONE

Dear Dr. Bar-Sela,

Thank you for submitting your manuscript to PLOS ONE. After careful consideration, we feel that it has merit but does not fully meet PLOS ONE’s publication criteria as it currently stands. Therefore, we invite you to submit a revised version of the manuscript that addresses the points raised during the review process.

Please address the issues raised.

We look forward to receiving your revised manuscript.

Kind regards,

Jeffrey Chalmers, Ph.D.

Academic Editor

PLOS ONE

Reviewers' comments:

Reviewer's Responses to Questions

**Comments to the Author**

1. If the authors have adequately addressed your comments raised in a previous round of review and you feel that this manuscript is now acceptable for publication, you may indicate that here to bypass the “Comments to the Author” section, enter your conflict of interest statement in the “Confidential to Editor” section, and submit your "Accept" recommendation.

Reviewer #1: (No Response)

Reviewer #2: All comments have been addressed

2. Is the manuscript technically sound, and do the data support the conclusions?

Reviewer #1: No

Reviewer #2: Yes

3. Has the statistical analysis been performed appropriately and rigorously? 

Reviewer #1: No

Reviewer #2: Yes

4. Have the authors made all data underlying the findings in their manuscript fully available?

Reviewer #1: No

Reviewer #2: Yes

5. Is the manuscript presented in an intelligible fashion and written in standard English?

Reviewer #1: Yes

Reviewer #2: No

6. Review Comments to the Author

Reviewer #1: The manuscript by Bar-Sela et al describes the assessment of extracellular vesicles (EVs) as biomarkers of endothelial injury in chemotherapy. The authors analyze cell-free plasma from patients with breast cancer (BC), colon cancer (CC) and healthy controls (HC) using NTA and flow cytometry, and also fractionate plasma using differential centrifugation to prepare a 20Kxg, 60’ pellet and analyze it by protein array and Western blot. The authors report increases in the total number of particles in plasma, and an apparent increase in size, as measured by NTA in CC patient plasma compared to controls. They also report changes in the expression of several membrane surface antigens as assessed by flow cytometry immunofluorescence. They identify angiostatin present in the 20Kxg pellet, as measured by protein array, as a marker of endothelial damage from chemotherapy.

Biomarkers of drug induced toxicity are needed to help manage patient treatment, and EVs are potentially useful. The observations reported in the present manuscript are aimed at developing such biomarkers, but inadequacies in the experimental methods used and omissions in the experimental design and reporting limit their interpretation.

The authors direct their efforts at cell-free plasma, which is a noble cause but its complexity challenges methods commonly used to analyze EVs. In the case of NTA, its limitations for the analysis of plasma are well documented and are related to its lack of specificity for EVs (it measures all particles that scatter light) and the confounding effects of lipoproteins, which are present at concentrations orders of magnitude higher than EVs. For these reasons, the NTA data (which are nearly impossible to evaluate due to the oddly scaled and poorly presented histograms that should be revised) should be presented with axes labeled as “Particles” rather than “EVs”. The authors should state what metric is being reported as Particle Size (mean, median, mode, etc), and what wavelength of laser light was used for illumination.

Similarly, the essential information and controls needed to interpret the flow cytometry data are lacking. These have been recently summarized in a position paper in the Journal of Extracellular Vesicles, and include information about the instrument operation and calibration and essential controls to demonstrate that the events being detected correspond to individual EVs. The laser and filter set up of the instrument should be reported according to the MIFlowCyt guidelines (a table in Supporting Info), the fluorescence results should be reported in MESF or antibodies/particle using the appropriate commercially available standards, and specificity controls including reagent-only, serial dilutions, and detergent treatment should be presented. It is recommended that the raw flow cytometry data files be submitted to a public repository (such as flowrepository.org). Where results are presented as % positive, the authors should include the denominator (the total concentration of particles/EVs) used to calculate that percentage, as well as the fluorescence threshold (reported in MESF or antibodies/event) used to determine positivity. The inclusion of these details will greatly improve the interpretability of the data.

The observation that angiostatin is increased in both the plasma 20kxg pellet and supernatant is interesting, but the authors contention that this is EV-encapsulated is speculative given the very crude nature and lack of characterization of these plasma preparations. Additional fractionation and characterization of the samples is required to speak to how much angiostatin is present in EVs vs free in plasma, as well as the nature of those EVs that might be carrying it.

Reviewer #2: Thank you for addressing concerns and questions.

Concerns have been addressed by the writers.

I have no additional concerns

7. PLOS authors have the option to publish the peer review history of their article (what does this mean?). If published, this will include your full peer review and any attached files.

Reviewer #1: No

Reviewer #2: No

---

## [Author Response · Author response to Decision Letter 1]

9 Sep 2020

After incorporating and implementing all the necessary corrections and adjustments suggested by the reviewers, we now believe that our work is suitable and ready for publication in the journal.

Specifically, we addressed the following issues:

1. Reviewer-1: In the case of NTA, its limitations for the analysis of plasma are well documented and are related to its lack of specificity for EVs (it measures all particles that scatter light) and the confounding effects of lipoproteins, which are present at concentrations orders of magnitude higher than EVs. 

We agree with the reviewer comment related to the NTA limitation; however, despite this limitations, the JOURNAL OF EXTRACELLULAR VESICLES (2018, VOL. 7, 1535750 https://doi.org/10.1080/20013078.2018.1535750) that serves as industry-standard states the" Minimal Information for Studies of Extracellular Vesicles 2018 (MISEV2018) state that "particle number can be measured by light scattering technologies, such as nanoparticle tracking analysis (NTA); by standard flow cytometry for larger EVs or high-resolution flow cytometry for smaller EVs" in our study we used both methods NTA and flow cytometry.

2. Reviewer-1: The NTA data should be presented with axes labeled as "Particles" rather than "EVs." The authors should state what metric is being reported as Particle Size (mean, median, mode, etc.), and what wavelength of laser light was used for illumination.

A new set of NTA figures in the higher resolution were attached (supplement figure 1). NTA data analysis methods are based on maximum-likelihood estimation, namely: finite track length adjustment (FTLA). The graph in the Supplement Figure-1 summarized 3-4 repeats of analysis for each sample and expressed as Averaged FTLA Concentration / Size

1. The following data were added to the method section, EVs size, and concentration:

"Camera Type – sCMOS. , Laser Module: NS300, 405nm; Software settings for analysis were kept constant for all measurements. Capture Settings: Camera Level: 13; Slider Shutter: 1232; Slider Gain: 219; FPS 25.0; Number of Frames: 749; Temperature: 25 oC, Viscosity: (Water) 0.86 cP; Syringe Pump Speed: 20 "

2. The figure legend was coerced as follow:

Supplement Figure-1: EVs size distribution diagram of a representative sample obtained from each study cohorts. The graphs summarized 3-4 repeats of analysis for each sample and expressed as Averaged FTLA Concentration / Size for HC-EVs (a), CC I-EVs (b), CC II-EVs (c), BC I-EVs (d), and BC II-EVs (e). 

1. The following Reference was added:

" Jamaly S, Ramberg C, Olsen R, Latysheva N, Webster P, Sovershaev T, Brækkan SK, Hansen JB. Impact of preanalytical conditions on plasma concentration and size distribution of extracellular vesicles using Nanoparticle Tracking Analysis. Sci Rep. 2018

3. Reviewer-1: Similarly, the essential information and controls needed to interpret the flow cytometry data are lacking. These have been recently summarized in a position paper in the Journal of Extracellular Vesicles and include information about the instrument operation and calibration and essential controls to demonstrate that the events being detected correspond to individual EVs. The laser and filter set up of the instrument should be reported according to the MIFlowCyt guidelines (a table in Supporting Info), the fluorescence results should be reported in MESF or antibodies/particle using the appropriate commercially available standards, and specificity controls including reagent-only, serial dilutions and detergent treatment should be presented. It is recommended that the raw flow cytometry data files be submitted to a public repository (such as flowrepository.org). Where results are presented as % positive, the authors should include the denominator (the total concentration of particles/EVs) used to calculate that percentage, as well as the fluorescence threshold (reported in MESF or antibodies/event) used to determine positivity. The inclusion of these details will greatly improve the interpretability of the data.

The manuscript was corrected according to the MIFlowCyt-EV recommendation (Joshua A. Welsh et al. JOURNAL OF EXTRACELLULAR VESICLES 2020, DOI: 10.1080/20013078.2020.1713526) 

1. The following information on instrument operation, calibration, and control were added to the methods section : 

• "PPP (50ul) were labeled with specific fluorescent antibodies for 30 minutes, at room temperature at dark, ended by adding of 450ul buffer FACS (DPBS, 0.02% Azid, 0.5% formaldehyde filtered by 0.22µm filter) without additional washing". 

• "Samples were analyzed by flow cytometers Canto II in log set at the voltage: FSC- 300, SSC-300, FITC-500, PE-350, APC-480. FSC threshold -700. Laser power: blue-20.39; red -12.24; violet-60. Unstained samples and controls were measured in the same setting (Supplement Figure 3)". 

• "Also, to validate that the particles analyzing by flow cytometer are phospholipid vesicles, detergent (1% tritonx100) was added to 50ul of PPP sample for 5 minutes in room temperature.". 

• The canto ii flow cytometer instrument cannot calculate total EVs concentration; this data obtained only by NTA.

 In addition, as a part of antibodies calibrations, D-PBS was labeled with several fluorescent antibodies, and events were collected during 1 minute at high speed. Part of the antibodies forms aggregates that can be seen in the R6 gate with a high autofluorescent rate. However, their number is less than 5% of the total EVs that analyzed in each PPP sample. Because we compare samples in the same conditions of labeling and FACS analysis, we can assume that if there are any artifacts related to the antibodies aggregates, it will affect all samples similarly.

2. The following information was added RESULTS section, EVs CELL ORIGINE :

"The majority of EVs measured by flow cytometer found to be a phospholipid vesicles located at the same area where 0.75µm beads are located (R6, Supplement Figure-1A), as confirmed by treatment of PPP with 1% triton-x100 that reduced EVs number in this area by above 10 times. (Supplement Figure 3)".

Legend of supplement figure 3 FACS analysis of PPP before and after triton treatment Unstained PPP EVs are located in the same area were 0.75µm beads are located (R6, Supplement Figure-1A). (A). HC11 - 79% EVs located at R6 (B). BC2 II: 92% EVs located at R6. Treatment with 1% triton-x100 reduced EVs number in the R6 area by above ten times. (C). HC 11: EVs located at R6, decreased to 6.6%. (D). BC2-II: EVs, located at R6, reduced to 8.8%. 

3. Reviewer-1: The observation that angiostatin is increased in both the plasma 20kxg pellet and supernatant is interesting, but the author's contention that this is EV-encapsulated is speculative given the very crude nature and lack of characterization of these plasma preparations. Additional fractionation and description of the samples is required to speak to how much angiostatin is present in EVs vs. free in plasma, as well as the nature of those EVs that might be carrying it.

Additional validation for the observation on EVs Angiostatin content was performed. WB validated plasma pellets of CC and BC patients and HC.

The additional section related to the WB was added to the methods as follow:

Western blot- "EV-pellets that were isolated from 500µl PPP were added to buffer lysis (x2, Ray biotech) with proteinase inhibitor 1% (Sigma). 50ug samples in buffer lysis were combined with 1:1 sample buffer with β-mercaptoethanol (1:20, Biorad). The samples were loaded and separated on mini protean TGX precast gel 4-20% and then transferred to the mini format of transfer blot turbo o.2um nitrocellulose membrane (Bio-Rad, Herculs CA, USA). This was followed by immunoblotting with rabbit polyclonal anti-human-Angiostatin (Abcam). After overnight incubation with the primary antibody, the membranes were washed and incubated with horseradish peroxidase (HRP) conjugated secondary antibodies (Cell Signaling Technology, Massachusetts, USA). Then, a chemiluminescence kit (WESTAR Nova 2, CYANAGEN, Bologna Italy) was used to detect the fluorescence. The western blot (WB) assay results were quantified using myECL™ Imager and analyzed using MyImageAnalysis Software (both from Thermo Fisher Scientific, Waltham, MA USA)".

In the results section- "WB analysis supports these findings; non-significant increase in Angistatin was found in EVs pellet obtained from patients at the end of chemotherapy in both diseases (CC II, BC II) compared to their levels before treatment (CC II, BC I)."

Gel images and graph were added to figure 6 (F, G), and the legend of figure 6 was amended: WB gel images of Angiostatin content in EVs pellet obtained from BC patients (n=4) and CC patients (n=4) in each time point (I, II) and in EVs of HC (E). Graph F summaries the WB results as average with STD of EVs Angiostatin as a ratio of Actin, normalized to HC.

---

## [Decision Letter · Decision Letter 2]

7 Oct 2020

Circulating Blood Extracellular Vesicles as a Tool to Assess Endothelial Injury and Chemotherapy Toxicity in Adjuvant Cancer Patients

PONE-D-20-06807R2

Dear Dr. Bar-Sela,

We’re pleased to inform you that your manuscript has been judged scientifically suitable for publication and will be formally accepted for publication once it meets all outstanding technical requirements.  Note, I was waiting for the other reviewer; however, I decided to go ahead with the one reviewers comments. 

Kind regards,

Jeffrey Chalmers, Ph.D.

Academic Editor

PLOS ONE

Additional Editor Comments (optional):

Reviewers' comments:

Reviewer's Responses to Questions

**Comments to the Author**

1. If the authors have adequately addressed your comments raised in a previous round of review and you feel that this manuscript is now acceptable for publication, you may indicate that here to bypass the “Comments to the Author” section, enter your conflict of interest statement in the “Confidential to Editor” section, and submit your "Accept" recommendation.

Reviewer #2: All comments have been addressed

2. Is the manuscript technically sound, and do the data support the conclusions?

Reviewer #2: Yes

3. Has the statistical analysis been performed appropriately and rigorously? 

Reviewer #2: N/A

4. Have the authors made all data underlying the findings in their manuscript fully available?

Reviewer #2: Yes

5. Is the manuscript presented in an intelligible fashion and written in standard English?

Reviewer #2: Yes

6. Review Comments to the Author

Reviewer #2: concerns are addressed

thank you for your submission

no other comments

best of luck with your future work

7. PLOS authors have the option to publish the peer review history of their article (what does this mean?). If published, this will include your full peer review and any attached files.

Reviewer #2: No

---

## [Editor Report · Acceptance letter]

14 Oct 2020

PONE-D-20-06807R2 

Circulating Blood Extracellular Vesicles as a Tool to Assess Endothelial Injury and Chemotherapy Toxicity in Adjuvant Cancer Patients 

Dear Dr. Bar-Sela:

I'm pleased to inform you that your manuscript has been deemed suitable for publication in PLOS ONE. Congratulations! Your manuscript is now with our production department. 

Kind regards, 

on behalf of

Dr. Jeffrey Chalmers 

Academic Editor

PLOS ONE